# The impact of art, storytelling, and STEAM-based approaches on creativity development in autistic youth and young adults: A mixed methods study protocol

**Jeana M. Holt** [1]*, **Katelyn Siekman**[2], **Margaret Fairbanks**[2], **Mark Fairbanks**[2], **Nathaniel Stern**[3]

**1** School of Nursing, University of Wisconsin-Milwaukee, Milwaukee, Wisconsin, United States of America,
**2** Islands of Brilliance, Milwaukee, Wisconsin, United States of America, **3** Peck School of the Arts and College of Engineering and Applied Science, University of Wisconsin-Milwaukee, Milwaukee, Wisconsin, United States of America

☯ These authors contributed equally to this work.
* jmholt@uwm.edu

**Data Availability Statement:** No datasets were generated or analyzed during the current study. All

## Abstract

There are contradicting perspectives regarding autistics' ability to be creative. Some researchers cite autistics' underlying social communication and interaction differences, fixated interests, and inflexible patterns as fundamentally inhibiting creativity. However, many autistics refute this mindset and produce creative works as painters, sculptors, photographers, and graphic artists. This protocol describes a mixed methods study that aims to determine the impact of art, storytelling, and STEAM-based approaches to develop autistic youth and young adult participants' creative self-efficacy, psychological empowerment, and design thinking traits. The primary research question is: How does a 120-minute workshop intervention impact the creative self-efficacy of autistic participants? We hypothesize that autistic participants' creative self-efficacy scores would linearly increase from the baseline measure. Exploratory research questions include: How does the workshop intervention affect the psychological empowerment and design thinking traits of the autistic participants? We hypothesize that autistic participants' psychological empowerment and design thinking traits scores would linearly increase from the baseline measure. We will use a design-based implementation research approach that values the collaboration between the researchers and educators who design interventions to understand when, how, and why learning happens. Recruitment began on 10 November 2023 and ended on 9 August 2024. The study's results are expected to be published in mid-2025. The study will provide crucial empirical evidence on the effects of an arts-based program on creative self-efficacy, psychological empowerment, and design thinking traits in autistic youth and young adults. We will use qualitative descriptive thematic analysis methods to analyze the digital images, stop motion videos, and participant presentations. Digital artifacts of the participants' creative process and the study team's analysis of the autistic youth's presentations will provide additional data to understand the study phenomenon's depth, meaning, and context. A mixed methods

relevant data from this study will be made available upon study completion.

**Funding:** This project is supported in part by the National Endowment for the Arts (NEA). JMH & NS received funding from the NEA, Award Number: 1925064-38. To find out more about how National Endowment for the Arts grants impact individuals and communities, visit www.arts.gov. The NEA did not play a role in the study design, data collection and analysis, decision to publish, or preparation of the manuscript.

**Competing interests:** The authors have declared that no competing interests exist.

design is advantageous when exploring complex issues that require empirical evidence and contextual understanding.

# Introduction

Depending on one's perspective, autism spectrum disorder (ASD) is seen as a deficit-laden disorder [1, 2] or a different way of interacting with the World that has positive outcomes [2–4]. The contradicting perspectives are also valid regarding autistics' ability to be creative [5–8]. Creativity is the ability to combine experiences, ideas, knowledge, and inspiration to create new things [9]. Some researchers cite the defining diagnostic criteria for ASD, i.e., underlying social communication and interaction deficits that persist across multiple settings with highly restricted, fixated interests and inflexible patterns [1], fundamentally inhibit creativity. One author reported that "autism imprisons its victims in a structure of restricted, repetitive behaviors and a diminished ability to uncover or accept new truths, interact socially, and communicate flexibly" (p. 273) [8]. However, many autistics refute this mindset and produce creative works as painters, sculptures, photographers, and graphic artists (e.g., [6, 10]. Furthermore, autistics may have an advantage in generating novel ideas (e.g., divergent thinkers) [11–13] compared to their neurotypical peers.

There is particular evidence of creative autistic works in their special interest area(s) [11, 14]. SpIn (Special + Interest) is the term that the autistic community developed to describe the areas of high interest that many autistics identify with throughout their lives [14]. SpIn is unique to the individual, e.g., plants, dates, sports, physics, and transportation [14]. Some researchers and educators view SpIn as restrictive and prohibitive to developing a creative mindset or abilities [15, 16]. However, a growing body of research promotes using SpIn as an entry point to creativity, education, and connections [14, 17]. Brown et al.[18] recently published the survey results of 1922 caregivers of autistic youth who identified creating and information seeking as adaptive activities that their youth engaged in regarding their SpIn. Educators reported autistic youth experience gains in fine motor skills, sensory precision, emotional understanding, executive control, and social and communication skills through SpIn activities [19]. In education, research results indicated that when autistic youth pursued their SpIn, they experienced relaxation, reduced anxiety, elevated happiness, and enhanced understanding of their physical surroundings [17]. Still, in its natal period [20], additional research is needed to understand the outcomes of incorporating SpIn into arts-based curricular development and community-based programming. Given the contradiction of the ASD diagnostic criteria and evidence of creativity in autistic people, additional research is needed to understand their creative capacity. We hypothesize that autistic youth who create a storyline using their SpIn will positively impact their creative self-efficacy, psychological empowerment, and design thinking traits.

## Islands of Brilliance

Islands of Brilliance is a non-profit organization that designs and implements creative programs for autistic and neurodivergent students ages eight through young adulthood. Specifically, the Islands of Brilliance uses "creativity, technology, and mentorship to positively change perceptions of self and promote broader community views of neurodivergent individuals as people with unlimited abilities rather than persons with disabilities" [21]. The Islands of Brilliance's educational model used the principles of project-based learning and universal design [22, 23], emphasizing participant-led creative self-expression and supporting the participant's

innate creativity. Islands of Brilliance believes in an individualized strengths-based rather than a deficit-based approach, which allows for unique creative expression and often results in outputs that look vastly different among participants. Thus far, evidence of success from programming at the Islands of Brilliance in augmenting these skills has been based on informal and internal surveys, and they have seen significant appreciation and growth.

Therefore, our planned mixed methods study will assess the impact of art, storytelling, and STEAM-based approaches to develop participants' creative self-efficacy, psychological empowerment, and design thinking in autistic youth and young adults (AYAs). This study is part of the National Endowment for the Arts (NEA)-funded research lab Autism Brilliance Lab for Entrepreneurship (ABLE) [24]. The primary research question is: How does the Sandbox@ workshop intervention impact the creative self-efficacy of autistic participants? We hypothesize that autistic participants' creative self-efficacy scores would linearly increase from the baseline measure. Exploratory research questions include: How does the Sandbox@ workshop intervention affect the psychological empowerment and design thinking traits of the autistic participants? We hypothesize that autistic participants' psychological empowerment and design thinking traits scores would linearly increase from the baseline measure.

## Materials and methods

### Study design

This study protocol details our planned mixed methods study, which will assess the impact of art, storytelling, and STEaM-based approaches to develop participants' creative self-efficacy [25, 26], psychological empowerment [27], and design thinking [28] in autistic AYAs. We will use a design-based implementation research (DBIR) approach that values the collaboration between the researchers and educators who design interventions to understand when, how, and why learning happens [29, 30]. The University of Wisconsin-Milwaukee's ethics review board determined this study is exempt.

### Sample and setting

The primary sample is autistic AYAs (8–30 years old). We chose this age group since most state-funded autism therapy services end in Wisconsin by age nine [31]. Therefore, there is a need for autism programs for this age group. Furthermore, developmental milestone progress in autism varies substantially, and some individuals may reach developmental milestones early, late, or not at all [32]. Therefore, chronological age does not always correlate with development [32]. The secondary sample is the parents or guardians (hereafter guardians) participating in the workshop. We will recruit a convenience sample of 30 autistic AYAs (8–30 years old). We will conduct the intervention in multiple community-based settings, e.g., museums, zoos, schools, and libraries in the Midwestern United States. An a priori power analysis was conducted using G*Power [33, 34] to determine the minimum sample size required to detect whether there are significant changes in creative self-efficacy (the primary outcome) over time. Results indicated that the required sample size to achieve 80% power for detecting a medium effect at a significance criterion of $\alpha = .05$ was $N = 20$ participants for ANOVA repeated measures within factors. Thus, the projected sample size of $N = 30$ will adequately test the study hypothesis.

### Inclusion criteria

Eligible participants in the study are AYAs who are (1) diagnosed with ASD according to the DSM V, (2) aged between 8 and 30 years old, (3) speak English, (4) willing and able to give

informed written consent or assent, (5) can respond to verbal instructions with visual supports to follow along with the structured activity, and (6) can complete a series of tasks independently or with facilitator or guardian support provided. These tasks include using physical art supplies (e.g., paper, pencils, glue, etc.) and/or iPads with drawing software and graphics editors to create a character, environment, storyline, and stop-motion video.

Eligible guardian participants are individuals who are (1) aged 18 years or older, (2) the parent or guardian of an autistic AYA enrolled in the program, (3) speak English, and (4) are willing and able to give informed written consent.

## Recruitment

We started recruiting participants for this study on the 10th of November 2023, through the Islands of Brilliance's website, email distribution list, conferences, and other workshops. The invitation to participate describes the study as "a study to help us understand better what aspects of Sandbox@ help students to improve their social, emotional, and technical skills." We use convenience sampling to enroll autistic AYAs who meet the inclusion criteria based on guardian reports. Recruitment will continue until the 9th of August 2024. We collect survey data using the university's Qualtrics system.

## Informed consent and assent

This study has several informed written assent/consent processes. Participants aged eight to 11 will provide written assent, and their guardians will provide written consent. Participants 12 and older will complete the combined written consent/assent forms with their guardians. Guardian participants will also complete a written consent form.

## Intervention: Sandbox@ workshop description

Authors MF and KS designed and facilitated the intervention. They hold graduate and doctoral degrees and certifications in education, special education, alternative education, bilingual education, and occupational therapy. The facilitators tailor the intervention to the participant's developmental needs, which are evaluated at least one week before the workshop during a 1:1 meeting with the family.

The facilitators designed the Sandbox@ intervention using the Situated learning theory (SLT) developed by Lave and Wenger [35]. SLT emphasizes the social and practical nature of learning, focusing on how people learn by becoming part of a community of practice and through active engagement with tasks and other learners. Learners start as novices and gradually move toward full participation by interacting with more experienced members (experts) who share knowledge, skills, and practices. In the SLT, learning occurs through social interaction, observation, and collaboration rather than formal instruction. Knowledge is co-created within the community and acquired through actively participating in experiences [35]. In autism research and practice, several aspects of SLT have been explored, including developing a community of practice where autistic individuals engage with neurotypical peers and other autistics in shared activities [36]. Other autism interventions use technology to practice social skills, decision-making, and problem-solving in a controlled yet contextually relevant environment [37].

The facilitators designed a 120-minute workshop to support autistic AYAs (8–30 years old). The workshop uses art and storytelling to develop participants' creative self-efficacy, psychological empowerment, and design thinking skills. Depending on their preference, participants work independently or within a group to ultimately produce a stop-motion video of a character, environment, and storyline they created. Participants can access various resources, from

physical art supplies to iPads with drawing software and graphics editors. Participants are asked to use their preferred materials to develop a character, environment, and storyline. Guardians and/or facilitators tailor their support to participants' needs.

First, the participant develops a character(s) in the Doodle Lounge activity. Workshop facilitators prompt participants with character development questions: "What do they do for work? Why do they live there? Do they have pets? Do they have family/friends?" After Doodle Lounge, participants complete a modified three-item creative self-efficacy instrument [25, 26] and upload an image or a short video describing their character.

Second, the participant creates the environment for the character(s) in the activity called Smactivity. Facilitators prompt participants to use their preferred supplies to create an environment for their animation. After Smactivity, participants complete a modified three-item creative self-efficacy instrument [25, 26] and upload an image or a short video describing their environment.

Third, the participant creates a stop motion video to tell a story in the Natterdays activity. Workshop facilitators direct participants to use the iPads and the Stop Motion Studios App to create a story with their character/s and environment. Facilitator prompts include: What happens? Does your character save the town from a villain? Does your character drive an ice cream truck through the town? You get to decide!! After Natterdays, participants can present their project to the group with the support of family or a facilitator. The facilitator prompts the participants to share their response to one of the following questions: What was the most fun thing you did today? What made you smile today? Participants choose how the group can celebrate their work, e.g., jazz hands, high five, applause, or no celebration. A research team member records the participant's presentation.

After the presentations, participants complete three instruments: modified three-item creative self-efficacy [25, 26], nine-item Psychological Empowerment (Menon, 2001), and ten-item Traits of Design Thinkers (Blizzard et al., 2015; Coleman et al., 2020). If needed, the facilitator will assist the participant in uploading their stop-motion video in Qualtrics. The study procedure is depicted in Fig 1.

## Procedure

**Data collection.** Table 1 summarizes the data collection times and measures of autistic AYA participants and their guardians. If the guardian and autistic AYA participants consent/assent online, a unique link will be sent to the guardian's email for the guardian and autistic AYA's consent/assent and pre-test survey. If the guardian and autistic AYA participant consent/assent immediately before the Sandbox@ workshop, they will scan a QR code to access their consent/assent and pre-test survey. We will provide iPads to complete the surveys immediately before and during the Sandbox@ workshop. At the end of each Sandbox@ workshop activity, the autistic AYA participant will use Qualtrics to complete the creative self-efficacy instrument and upload a video, audio, or screenshot of their progress. We will also video record the autistic AYA participant's presentation to the group. All surveys (see Table 1) are completed online via the web application Qualtrics.

To de-identify the data from participants, the research team will create a key linking their names to their pseudonyms, response types, whether they are identified as neurodiverse or neurotypical, and survey data. The research team will also create a key link between guardians' names and their pseudonyms, emails, and survey data.

**Measures.** The _Creative Self-efficacy (CSE)_ instrument [25, 26] assesses a person's perceived capacity for creative activities. The CSE instrument has three items, measured on a scale ranging from 1 (strongly disagree) to 5 (strongly agree). Tierney and Farmer [25]

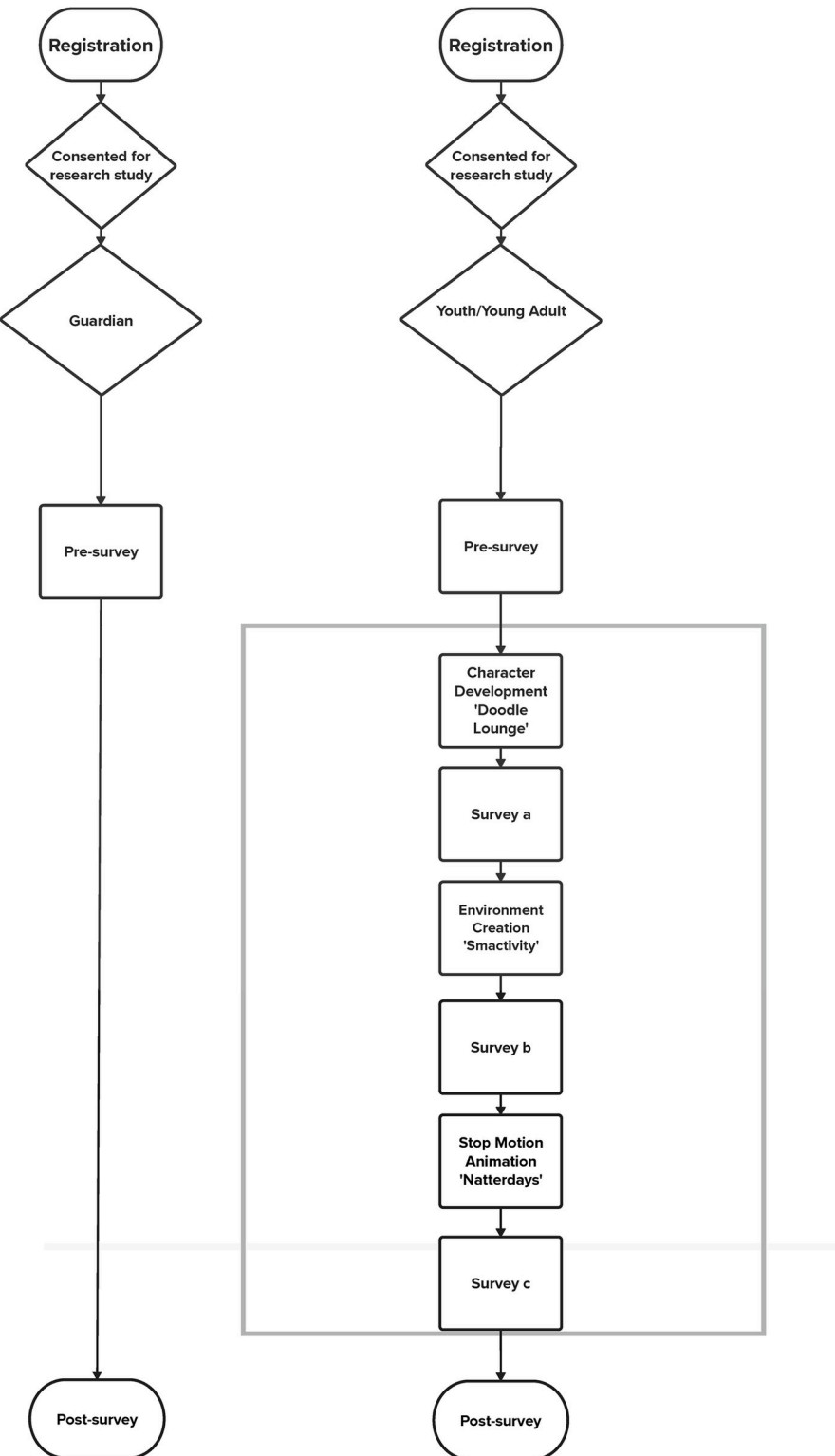

**Fig 1. Study procedure.** This figure provides a step-by-step workflow for the study. (A) AYA participants and their guardians were recruited based on inclusion and exclusion criteria. (B) AYA participant data were collected during three main phases: pre-intervention baseline, intervention, and post-intervention follow-up. (C) Guardian participant data were collected at: pre-intervention baseline and post-intervention follow-up.

**Table 1. Summary of measures and data collection timeline.**

| Measurement Tools | Pre-test Youth | Pre-test Adult | Survey A Youth | Survey A Adult | Survey B Youth | Survey B Adult | Survey C Youth | Survey C Adult | Post-test Youth | Post-test Adult |
|---|---|---|---|---|---|---|---|---|---|---|
| Creative Self-Efficacy (20) | X | X | X | | X | | X | | X | X |
| Psychological Empowerment [22] | X | X | | | | | X | | X | X |
| Traits of Design Thinkers [23, 30] | X | X | | | | | X | | X | X |
| File upload of character development via screenshot or video | | | X | | | | | | | |
| File upload of environment development via screenshot or video | | | | | X | | | | | |
| File upload of Stop motion animation video | | | | | | | X | | | |
| Video recording of AYA presentation | | | | | | | X | | | |
| Demographics | | | | | | | | | | X |
| Neuro characteristics | | | | | | | | | | X |
| Focused Interests [12] | | | | | | | | | | X |

Pre-test: administered immediately before Sandbox@

Sandbox@ Survey A: administered after Doodle Lounge

Sandbox@ Survey B: administered after Smactivity

Sandbox@ Survey C: administered after presentations

Post-test: administered one week after Sandbox@

reported good reliability ranging from .83 to .87 and acceptable test-retest reliabilities, Time 1 $\alpha$ = .74; Time 2 $\alpha$ = .81 [26]. The measure has been used in employee and school-based settings. Tierney and Farmer [25] reported that CSE was positively related to within-person creative performance improvement when tested in a work context, which supports its construct validity. Factor analyses conducted in the original study [25] and later research [26] have also supported the unidimensional structure of CSE, further indicating construct validity. Study results indicate that middle and high school students reported higher levels of CSE after instructors gave them positive feedback and encouragement regarding their imagination [38].

*Psychological Empowerment (PE)* is described as a sequence of actions that lead to an empowered state [27]. The PE instrument is a nine-item self-report instrument using a 6-point Likert Scale (1: strongly disagree to 6: strongly agree). Three sub-scales measure the main dimensions of Psychological Empowerment: (a) perceived control, (b) perceived competence, and (c) goal internalization. PE has high internal consistency, with Cronbach's alpha values ranging from 0.80 to 0.90 for the overall scale and its subscales [27]. Initially tested in college students, the instrument had acceptable test-retest reliabilities: goal internalization (.86), perceived control (.87), and perceived competence (.77). A subsequent validation study in employees of a financial organization yielded similar results: goal internalization (.86), perceived control (.86), and perceived competence (.78) [27]. Its predictive validity has been established in various organizational settings, suggesting that employees' empowerment perceptions measured by this instrument can reliably predict critical work-related outcomes such as job satisfaction and employee commitment [39]. A recent study of college students [40] reported that an in-person classroom exercise where students experienced being empowered and disempowered fostered structural and psychological empowerment and provided preliminary evidence on the outcomes of a 90-minute empowerment intervention.

*Traits of Design Thinkers* (TDT) encompass several domains, including curiosity, persistence, divergent thinking, and resilience. Design thinkers also embrace novel problem-solving methods [28, 41]. TDT is a 10-item instrument using a 5-point Likert Scale (0: strongly disagree to 4: strongly agree). Five sub-scales measure the following traits: feedback seeking, integrative thinking, optimism, experimentalism, and collaboration. This instrument had an

acceptable level of reliability in undergraduate students (Cronbach's alpha = 0.76). Studies using the TDT instrument report good internal consistency, with Cronbach's alpha values typically in the range of 0.70 to 0.90 across different dimensions of design thinking traits [28, 42]. This suggests that the items within each subscale reliably measure their respective traits. Coleman and colleagues [28] suggest that the TDT instrument can understand students' baseline design thinking traits and tailor programming and curricula to foster design thinking abilities.

**Cognitive interviews.** The measures in the study were psychometrically tested in diverse samples; however, the authors did not explicitly test their reliability and validity in autistic participants. Therefore, a research team member with qualitative research expertise conducted cognitive interviews to understand the meanings and processes participants may use to answer the questions. The research team member interviewed three guardians, five facilitators, and two autistic AYAs who had previously participated in programs offered by the Islands of Brilliance. The interviews were tailored based on the interviewee. This research team member used the cognitive interview technique of verbal probing [43, 44] to elicit information relevant to revising the three instruments for autistic participants. First, the interviewer read a survey item aloud and then asked probing questions. Then, different probes [43, 44] were used after reading each item to enhance the validity of the conclusions the research team will draw from the revised instruments. The following is an example of the process. The research team read the Item: I have confidence in solving problems creatively. Then asked,

- What does this statement mean to you? How do you think a person with autism may interpret this statement? (Comprehensive/Interpretation Probes)

- How would you reword it to make more sense to you? How would you reword the statement to make more sense to someone with autism? (Paraphrasing Probes)

- Do you find this question easy or difficult to answer? Do you think that a person with autism would find this item easy or difficult to answer? (General probes)

Based on several rounds of feedback from the interviewees, the research team member iteratively revised the instruments. Then, the research team member presented the revised instruments to the entire research team and Islands of Brilliance leaders, who reached a consensus on a final version of each instrument that best met the anticipated verbal and cognitive needs of the autistic AYA participants. Table 2 presents the original and revised items and instruments for the study.

## Data management

The research team will use Qualtrics for quantitative data management. This secure, web-based application is designed to support research data collection. It provides data validation, audit trails, and automated export procedures to various statistical packages. The qualitative data (videos, audio, screenshots, and presentations) will be managed using QSR NVivo software [45]. All media files and transcripts uploaded to NVivo are encrypted while in storage and transit.

**Data monitoring.** The data monitoring committee comprises the study team, the Islands of Brilliance leadership, and the University of Wisconsin-Milwaukee's Institutional Review Board (IRB). This study is low-risk, yet several monitoring processes are in place to protect the participants. Autistic AYA and guardian participants are provided with the study team's phone number, email address, and the contact phone numbers of members of the IRB. They can notify the principal investigator of any harm, which will be reported to the local institution. The study team will not conduct interim analyses because of the low-risk nature of the trial.

**Table 2. Original and revised items in instruments after cognitive interviews.**

| Creative Self-efficacy 7-point Likert Scale 1, very strongly disagree to 7, very strongly agree. | | | | |
|---|---|---|---|---|
| **Item** | **Scale** | **Sub-scale** | **Original Item** | **Revised Item** |
| 1 | Creative self-efficacy | | I have confidence in my ability to solve problems creatively. | I can imagine new ways of doing things. |
| 2 | Creative self-efficacy | | I have a knack for further developing the ideas of others. | I am good at adding to others' ideas. |
| 3 | Creative self-efficacy | | I feel that I am good at generating novel ideas. | I am good at thinking of new ideas. |
| **Psychological Empowerment** 6-point Likert Scale (1 strongly disagree, to 6 strongly agree) | | | | |
| Item | Scale | Sub-scale | Original Item | Revised Item |
| 1 | Empowerment | Goal Internalization | I am inspired by what we are trying to achieve as an organization. | I am excited about what the project will look like at the end |
| 2 | Empowerment | Goal Internalization | I am inspired by the goals of the organization. | I am excited by the activities in the project. |
| 3 | Empowerment | Goal Internalization | I am enthusiastic about working toward the organization's objectives. | I am excited about doing new projects. |
| 4 | Empowerment | Perceived Control | I can influence the way work is done in my department. | I have a say in my future projects. |
| 5 | Empowerment | Perceived Control | I can influence decisions taken in my department. | I can change my environment. |
| 6 | Empowerment | Perceived Control | I have the authority to make decisions at work. | I have the power to make decisions. |
| 7 | Empowerment | Perceived Competence | I have the capabilities required to do my job well. | I have the skills to do my project well. |
| 8 | Empowerment | Perceived Competence | I have the skills and abilities to do my job | I can do a good job on my project. |
| 9 | Empowerment | Perceived Competence | I have the competence to work effectively. | I can work well. |
| **Traits of Design Thinkers** 5-point Likert Scale (0-Strongly disagree" to "4-Strongly agree) | | | | |
| Item | Scale | Sub-scale | Original Item | Revised Item |
| 1 | Design Thinking Traits | Feedback Seeking | I understand the value of seeking feedback while my work is in progress. | I like asking for help on a project. |
| 2 | Design Thinking Traits | Feedback Seeking | I think seeking input from the user is an important part of the design process. | I think asking for help is an important part of the project. |
| 3 | Design Thinking Traits | Integrative Thinking | I think about how solutions can be integrated into a larger context. | I can add new ideas to a project. |
| 4 | Design Thinking Traits | Integrative Thinking | I like to narrow my focus when deriving solutions. | I can focus on different steps toward a goal |
| 5 | Design Thinking Traits | Optimism | I view challenges as opportunities, not as threats. | I can learn from challenges. |
| 6 | Design Thinking Traits | Optimism | My attitude worsens when a problem is challenging to solve. | I get overwhelmed when a problem becomes hard. |
| 7 | Design Thinking Traits | Experimentalism | I ask questions when searching for new approaches to problem-solving. | I ask questions when exploring new ideas to solve problems. |
| 8 | Design Thinking Traits | Experimentalism | Generating ideas is an important part of the design process. | I can think of new ideas for my project. |
| 9 | Design Thinking Traits | Collaboration | I value the perspectives of my teammates when working in a group. | I like to hear ideas from my peers. |
| 10 | Design Thinking Traits | Collaboration | I find collaborative work is a waste of time. | I find group work a waste of time. |

**Ethical considerations.** The University of Wisconsin-Milwaukee's IRB approved the study before enrollment. The study PI will report any changes to the protocol to the IRB and all study team members. All study personnel have completed training on protecting human subjects in research. Data will be stored on a secure server with physical, technical, and administrative access controls. Remote access is available over a secure network via encrypted connections to password-authorized users. Files with participant identifiers will be stripped of identifiers as soon as they are no longer needed. The study staff have no conflicts of interest.

## Analysis

**Quantitative analysis.**   To address the research questions, linear mixed-effects models will be used to analyze the repeated measurements of the outcomes of interest. Specifically, the change in the outcomes across the measurement times from baseline will be explicitly tested while controlling for the potential effects of age and sex.

**Exploratory factor analysis.**   Exploratory factor analysis (EFA) will be performed using the maximum likelihood method for factor extraction with varimax rotation. The psych R package will be used on the pre-test CSE [25, 26], PE [27], and TDT [28] instruments. The EFA analyses will be conducted to validate the instruments used for this study in the autistic AYA sample. The R programming language (version 4.3.1) will be used for data management and all analyses (R Core Team, 2023). The supporting R packages include dyplyr (version 1.1.3), magrittr (2.0.3), foreign (version 0.8–85), Matrix (version 1.6–1.1), mice (3.16.0), and lme4 (version 1.1–34). Statistical tests will be considered significant at the $\alpha$ level $< 0.05$.

**Qualitative analysis.**   We will use established qualitative descriptive thematic analysis methods to analyze the digital images, stop motion videos, and participant presentations [46, 47]. Specifically, the inductive thematic analysis will include the following systematic process: (1) initialization—reading, highlighting, and coding transcriptions and viewing the digital data to identify dimensions of the participants' data; (2) construction—classifying, comparing, labeling, translating, and defining and describing the codes and themes (p. 105, [47]; (3) rectification—verifying the appraisal and analysis process thus far to assess for completeness; and (4) finalization—creating a "storyline" that details the process from raw data to developing a holistic story of the study phenomenon [46, 47]. During the process, we will employ research team debriefing, reflexive journaling, and member checking [48] to increase the trustworthiness of the data by verifying preliminary results with autistic participants [3].

## Dissemination

We intend to write and publish two manuscripts. The first corresponds to the quantitative findings and exploratory factor analysis (EFA) of the creative self-efficacy [25, 26], psychological empowerment [27], and traits of design thinkers [28] instruments. The second corresponds to the qualitative findings from analyzing the autistic AYA's digital artifacts and presentations. Additionally, we will share the study results in presentations, videos, blog posts, and publications to academic, public, and neurodiverse audiences.

## Discussion

Our study is designed to examine the effect of a STEAM-based program involving storytelling on the creative self-efficacy, psychological empowerment, and design thinking traits of autistic participants. We will use a longitudinal mixed methods design, which measures key constructs over time to detect trends [49]. Embedded in this study is a high degree of collaborative and systematic inquiry where we equally respect the researcher's and practitioner's expertise. These tenets are highly valued in design-based implementation research [29, 30]. We will repeatedly measure creative self-efficacy, psychological empowerment, and traits of design thinkers before, during, and after the creative process. The instruments have all been shown to be valid and reliable tools for employees [27] and middle, high school, and university students [38, 40]. It is unknown how reliable the modified tools will be in detecting a change in the perceived capacity for creativity, psychological empowerment, or design thinking traits of the autistic AYA participants. Therefore, we are also examining digital artifacts during the creative process and our direct observations of the autistic AYA's presentations to provide additional data to

develop a holistic story [47] of the study phenomenon and main outcome, creative self-efficacy.

There is preliminary evidence of creativity in autistic AYAs [2, 4, 7]. However, a recent systematic review and meta-analysis reported reduced creative output fluency, flexibility, and representation in creative fields [7]. It is essential to remember that ASD is a specific diagnosis, but autistic individuals are incredibly unique and diverse [50]. There continues to be a gap in the ideal ways to measure creativity in autistic AYAs and research reporting what creative output looks like in this population [50]. Our study more closely aligns with the art therapy perspective, where the creative process is more important than the creative output [51]. We hypothesize that empowering autistic AYAs to create a character, environment, and storyline using their SpIn will positively impact their creative self-efficacy. Further, we view the creative process as unique to be measured within participants versus across participants.

The study will fill significant research gaps in the community-based autistic AYA research and arts-based curricular development that uses SLT and asset-based view of participants instead of focusing on deficits or areas of weakness [52]. Unfortunately, some autism researchers continue to use a deficit framework, concentrating on a disorder that needs to be cured versus an identity with positive characteristics [53]. The study is designed to examine within participant change, acknowledging that there is a high degree of heterogeneity among autistic AYAs [7]. The results from this study will lead to a more comprehensive understanding of if and how autistic AYAs' creative capacity changes when they learn in a supportive and empowering environment that celebrates their unique abilities and interests.

## Limitations

We acknowledge the following limitations of this protocol and present contingency strategies. This research study will occur within midwestern communities, which may limit the generalizability of the results to other geographic areas with different resource availability [54, 55]. Further, the convenience sampling recruitment strategy might lead to participants who differ from the broader population. A non-randomized sample may also lead to self-selection bias, where individuals who choose to participate have differing beliefs or values from the wider population. To mitigate those limitations, we will recruit participants using social media and from various school and community-based settings, conferences, and rural and urban communities to diversify our sample. In previous work, we have successfully employed several recommended strategies [56–58] to recruit autistic AYA participants, including using the Islands of Brilliance's public engagement. This organization's media campaigns garnered nearly 35 million impressions. They also promoted our collaborative work at 130 outreach events in 18 months.

We will ensure the anonymity and confidentiality of all participants to reduce consent bias. This bias may occur if, at the time of consent, participants differ from the general autistic population [59, 60]. The participants in this study will be limited to those individuals who can communicate by speaking in English. This inclusion criterion excludes non-speaking autistic people, limiting the findings' generalizability. Finally, there may be challenges in interpreting the quantitative and qualitative results. We will limit those challenges by sharing preliminary findings with study participants to verify the accuracy and validity through member checking [61], a method endorsed by the autism community [3].

## Conclusion

The study's quantitative outcomes will provide crucial empirical evidence on the effects of an arts-based program on creative self-efficacy, psychological empowerment, and design thinker

traits in autistic AYAs. Digital artifacts of the participants' creative process and analysis of the autistic AYA's presentations will provide additional qualitative data to explore the study phenomenon's depth, meaning, and context. The mixed methods study leverages the strengths of each paradigm, which is advantageous when exploring nuanced issues that require empirical evidence and contextual understanding [47, 61].

## Author Contributions

**Conceptualization:** Jeana M. Holt, Katelyn Siekman, Margaret Fairbanks, Mark Fairbanks, Nathaniel Stern.

**Data curation:** Jeana M. Holt, Nathaniel Stern.

**Formal analysis:** Jeana M. Holt.

**Funding acquisition:** Jeana M. Holt, Nathaniel Stern.

**Investigation:** Jeana M. Holt, Nathaniel Stern.

**Methodology:** Jeana M. Holt, Nathaniel Stern.

**Project administration:** Jeana M. Holt, Nathaniel Stern.

**Resources:** Jeana M. Holt.

**Supervision:** Jeana M. Holt, Nathaniel Stern.

**Validation:** Jeana M. Holt, Katelyn Siekman, Margaret Fairbanks, Mark Fairbanks, Nathaniel Stern.

**Writing – original draft:** Jeana M. Holt.

**Writing – review & editing:** Jeana M. Holt, Katelyn Siekman, Margaret Fairbanks, Mark Fairbanks, Nathaniel Stern.

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
