## [Decision Letter · Decision Letter 0]

2 Sep 2024

PONE-D-24-28708The impact of art, storytelling, and STEAM-based approaches on creativity development in autistic youth and young adults: A mixed methods study protocolPLOS ONE

Dear Dr. Holt,

Thank you for submitting your manuscript to PLOS ONE. After careful consideration, we feel that it has merit but does not fully meet PLOS ONE’s publication criteria as it currently stands. Therefore, we invite you to submit a revised version of the manuscript that addresses the points raised during the review process.

We look forward to receiving your revised manuscript.

Kind regards,

Cho Lee Wong, PhD

Academic Editor

PLOS ONE

Journal Requirements:

 1. When submitting your revision, we need you to address these additional requirements. Please ensure that your manuscript meets PLOS ONE's style requirements, including those for file naming. The PLOS ONE style templates can be found at https://journals.plos.org/plosone/s/file?id=wjVg/PLOSOne_formatting_sample_main_body.pdf and https://journals.plos.org/plosone/s/file?id=ba62/PLOSOne_formatting_sample_title_authors_affiliations.pdf. 2. When completing the data availability statement of the submission form, you indicated that you will make your data available on acceptance. We strongly recommend all authors decide on a data sharing plan before acceptance, as the process can be lengthy and hold up publication timelines. Please note that, though access restrictions are acceptable now, your entire data will need to be made freely accessible if your manuscript is accepted for publication. This policy applies to all data except where public deposition would breach compliance with the protocol approved by your research ethics board. If you are unable to adhere to our open data policy, please kindly revise your statement to explain your reasoning and we will seek the editor's input on an exemption. Please be assured that, once you have provided your new statement, the assessment of your exemption will not hold up the peer review process. 3. Please include your full ethics statement in the ‘Methods’ section of your manuscript file. In your statement, please include the full name of the IRB or ethics committee who approved or waived your study, as well as whether or not you obtained informed written or verbal consent. If consent was waived for your study, please include this information in your statement as well. 

Additional Editor Comments:

This mixed-methods protocol aimed to determine the impact of art, storytelling, and STEAM-based approaches on developing autistic youth and young adult participants' creative self-efficacy, psychological empowerment, and design thinking traits. The topic of this study is interesting and important. However, the rationale for this study and the theoretical/ conceptual framework for the intervention development require further clarification. Below are some comments for consideration:

Introduction:

1. Has the Spln approach been used with the target population before? Please justify why the proposed approach has the potential to improve self-efficacy, empowerment and design thinking. What are the hypotheses behind it?

Sampling:

1. Why choose 8-30 years old?

2. Please revisit the calculation method of sample size.

Inclusion criteria:

1. Please describe what a "series of tasks independently" is.

Intervention:

1. What theory or frameworks guide the development of the intervention?

2. Who delivers the intervention? What qualifications does he/she have?

3. Are there differences in the delivery of interventions due to the wide age range?

Measures:

1. Have all outcome measures been used and tested in autistic youth and young adults? What are their reliability and validity?

Reviewers' comments:

Reviewer's Responses to Questions

**Comments to the Author**

1. Does the manuscript provide a valid rationale for the proposed study, with clearly identified and justified research questions?

Reviewer #1: Yes

Reviewer #2: Yes

2. Is the protocol technically sound and planned in a manner that will lead to a meaningful outcome and allow testing the stated hypotheses?

Reviewer #1: Yes

Reviewer #2: Yes

3. Is the methodology feasible and described in sufficient detail to allow the work to be replicable?

Reviewer #1: No

Reviewer #2: Yes

4. Have the authors described where all data underlying the findings will be made available when the study is complete?

Reviewer #1: No

Reviewer #2: Yes

5. Is the manuscript presented in an intelligible fashion and written in standard English?

Reviewer #1: Yes

Reviewer #2: Yes

6. Review Comments to the Author

You may also provide optional suggestions and comments to authors that they might find helpful in planning their study.

Reviewer #1: The title and abstract of this article are strong. The title clearly indicates that the study focuses on enhancing creativity in autistic youth through art and storytelling. The abstract provides a concise summary of the research goals, methods, and expected outcomes. However, it lacks specific details about the outcomes that will be measured, which could improve reader comprehension. Overall, the article shows promise, and addressing this aspect could enhance its effectiveness.

The introduction effectively presents a comprehensive background on the differing opinions regarding creativity in autistic individuals, establishing a solid foundation for the research. It clearly outlines the research questions and hypotheses that arise from this context. The significance of the study is well-articulated, highlighting the need for further exploration. However, incorporating more recent studies would enhance the introduction's relevance and situate it within the current literature.

The study employs a mixed methods design, which aligns well with its objectives and allows for a thorough investigation of creativity in autistic youth. The data collection methods are clearly defined, and the sample size of 30 participants is justified, adding to the study's credibility. However, the methodology section could be strengthened by detailing the participant recruitment process to address potential biases. Additionally, providing more specific information on the validity and reliability of the instruments used would further enhance the methodology.

The methodology demonstrates a clear approach to data presentation and analysis, utilizing mixed methods for a comprehensive understanding. However, the lack of preliminary results or discussion of potential challenges in data interpretation leaves readers without insight into expected outcomes, limiting their understanding of the study's potential findings and impact.

The forthcoming discussion section is anticipated to enrich the study by connecting findings to existing literature and exploring the implications for understanding creativity in autistic individuals. The expected acknowledgment of limitations and suggestions for future research reflects a thoughtful approach. However, the absence of the discussion section currently limits a complete evaluation of the study's conclusions and overall impact.

The conclusion succinctly summarizes the study's objectives and potential contributions, reinforcing its focus by reiterating the research questions and hypotheses. However, due to the lack of results, the conclusion does not provide concrete insights or evidence-based recommendations, which limits its practical applications.

This innovative study explores the relationship between creativity and autism through arts-based methods, holding significant potential for the field. The anticipated findings could transform our understanding of the creative capabilities of autistic individuals, influencing educational practices and challenging existing perceptions of autism. However, to fully highlight its contributions, the study needs to establish clearer connections to existing research, clarifying how it builds upon or diverges from prior work.

The article presents a well-structured and relevant study protocol that is likely to contribute significantly to the understanding of creativity in autistic youth and young adults. While it has several strengths, including a clear research design and a solid theoretical background, there are also areas for improvement, particularly in providing more detailed methodological information and addressing potential biases in participant recruitment. The anticipated results and discussions will ultimately determine the study's impact on the field.

Reviewer #2: ASD was diagnosed according to DSM ?

It has been mentioned at the end of abstract that multiple paradigms will be used, so mention which multiple paradigms will be used?

Any training obtained for the intervention?

qualitative analysis was done through which software?

7. PLOS authors have the option to publish the peer review history of their article (what does this mean?). If published, this will include your full peer review and any attached files.

Reviewer #1: **Yes: **Dr. Syeda Rubab Aftab

Reviewer #2: No

---

## [Author Response · Author response to Decision Letter 0]

20 Oct 2024

PONE-D-24-28708

The impact of art, storytelling, and STEAM-based approaches on creativity development in autistic youth and young adults: A mixed methods study protocol

PLOS ONE

Cho Lee Wong, PhD

Academic Editor

PLOS ONE

Dear Dr. Wong,

We appreciate the opportunity to revise our manuscript based on your and the reviewers’ comments and recommendations. Our submission has enhanced accuracy, clarity, and quality. We responded to each point you and the reviewers raised.

Kind regards,

Dr. Holt

● Thank you for sharing the links. We revised the submission adhering to the templates. 

● Thank you for these directions. We will revise the statement during the submission process.

● Thank you for asking for clarifications on those in the Methods section. 

○ We added the full name of the institution’s IRB: University of Wisconsin-Milwaukee.

○ We added that we obtained informed written assent and consent.

Additional Editor Comments:

This mixed-methods protocol aimed to determine the impact of art, storytelling, and STEAM-based approaches on developing autistic youth and young adult participants' creative self-efficacy, psychological empowerment, and design thinking traits. The topic of this study is interesting and important. However, the rationale for this study and the theoretical/ conceptual framework for the intervention development require further clarification. Below are some comments for consideration:

Introduction:

1. Has the Spln approach been used with the target population before? Please justify why the proposed approach has the potential to improve self-efficacy, empowerment and design thinking. What are the hypotheses behind it? 

● We appreciate the opportunity to expand on these topics. We expanded the introduction to include recent research findings when researchers and educators included SpIn as the focus of activities in educational settings. Those findings support our hypotheses, which we added before we describe the Islands of Brilliance. 

Sampling:

1. Why choose 8-30 years old?

● This is an excellent question. We chose the sample range based on a couple of factors. We added the following text to the sample and setting section to clarify this for the readers.

● The primary sample is autistic AYAs (8-30 years old). We chose this age group since most state-funded autism therapy services end in Wisconsin by age nine. Therefore, there is a need for autism programs for this age group. Furthermore, developmental milestone progress in autism varies substantially, and some individuals may reach developmental milestones early, late, or not at all. Therefore, chronological age does not always correlate with development (Kuo et al., 2022).

2. Please revisit the calculation method of sample size.

● Thank you for this suggestion. We recalculated the sample size with one group to determine an a priori sample size of 20. We revised the text accordingly. 

1. Please describe what a "series of tasks independently" is.

● We appreciate the clarifying question. We added the following text to the inclusion criteria.

● (5) can complete a series of tasks independently or with facilitator or guardian support provided. These tasks include using physical art supplies (e.g., paper, pencils, glue, etc.) and/or iPads with drawing software and graphics editors to create a character, environment, storyline, and stop-motion video.

Intervention:

1. What theory or frameworks guide the development of the intervention?

● Thank you for the opportunity and suggestion to include this information. We have added the following paragraph to the Intervention: Sandbox@ workshop description section.

● The Sandbox@ intervention was designed using the Situated learning theory (SLT) developed by Lave and Wenger (1991). SLT emphasizes the social and practical nature of learning, focusing on how people learn by becoming part of a community of practice and through active engagement with tasks and other learners. Learners start as novices and gradually move toward full participation by interacting with more experienced members (experts) who share knowledge, skills, and practices. Learning occurs through social interaction, observation, and collaboration rather than formal instruction. Knowledge is co-created within the community; it is implicit or tacit, acquired through actively participating in experiences (Lave & Wenger, 1991). In autism research and practice, several aspects of SLT have been explored, including developing a community of practice where autistic individuals engage with neurotypical peers and other autistics in shared activities (Gillberg et al. 2024). Other autism interventions use technology to practice social skills, decision-making, and problem-solving in a controlled yet contextually relevant environment (Zhang et al. 2022).

2. Who delivers the intervention? What qualifications does he/she have?

● Thank you for this suggestion. The intervention description begins with who designed and facilitated the intervention. 

● Authors MF and KS designed and facilitated the intervention. They hold graduate and doctoral degrees and certifications in education, special education, alternative education, bilingual education, and occupational therapy. The facilitators tailor the intervention to the participant's needs, which are evaluated at least one week before the workshop during 1:1 meetings with the family.

3. Are there differences in the delivery of interventions due to the wide age range? 

● We appreciate the ability to clarify this point. We added the following text to the sampling section. 

● Furthermore, developmental milestone progress in autism varies substantially, and some individuals may reach developmental milestones early, late, or not at all. Therefore, chronological age does not always correlate with development (Kuo et al., 2022).

Measures:

1. Have all outcome measures been used and tested in autistic youth and young adults? What are their reliability and validity? (Design thinking autism abstracts for DT articles)

● The measures were not explicitly used in autistic youth and young adults; therefore, we conducted cognitive interviews to understand participants' meanings and processes for answering questions. We added the following text to the Cognitive Interview section to clarify the intention of the interviews. 

● The measures in the study were psychometrically tested in diverse samples; however, the authors did not explicitly test their reliability and validity in autistic participants. Therefore, a research team member with qualitative research expertise conducted cognitive interviews to understand meanings and processes participants may use to answer questions. The research team member interviewed three guardians, five facilitators, and two autistic AYAs who had previously participated in programs offered by the Islands of Brilliance.

Reviewers' comments:

Reviewer's Responses to Questions

Comments to the Author

1. Does the manuscript provide a valid rationale for the proposed study, with clearly identified and justified research questions?

Reviewer #1: Yes

Reviewer #2: Yes

2. Is the protocol technically sound and planned in a manner that will lead to a meaningful outcome and allow testing the stated hypotheses?

Reviewer #1: Yes

Reviewer #2: Yes

3. Is the methodology feasible and described in sufficient detail to allow the work to be replicable?

Reviewer #1: No

Reviewer #2: Yes

4. Have the authors described where all data underlying the findings will be made available when the study is complete?

Reviewer #1: No

Reviewer #2: Yes

5. Is the manuscript presented in an intelligible fashion and written in standard English?

Reviewer #1: Yes

Reviewer #2: Yes

6. Review Comments to the Author

You may also provide optional suggestions and comments to authors that they might find helpful in planning their study.

Reviewer #1: The title and abstract of this article are strong. The title clearly indicates that the study focuses on enhancing creativity in autistic youth through art and storytelling. The abstract provides a concise summary of the research goals, methods, and expected outcomes. However, it lacks specific details about the outcomes that will be measured, which could improve reader comprehension. Overall, the article shows promise, and addressing this aspect could enhance its effectiveness.

● Thank you for sharing these opportunities for improvement. We added content about the outcome measures: creative self-efficacy, psychological empowerment, and traits of design thinkers in the text. 

The introduction effectively presents a comprehensive background on the differing opinions regarding creativity in autistic individuals, establishing a solid foundation for the research. It clearly outlines the research questions and hypotheses that arise from this context. The significance of the study is well-articulated, highlighting the need for further exploration. However, incorporating more recent studies would enhance the introduction's relevance and situate it within the current literature.

● Thank you for sharing these points. We updated the introduction to include references to more recent studies. 

The study employs a mixed methods design, which aligns well with its objectives and allows for a thorough investigation of creativity in autistic youth. The data collection methods are clearly defined, and the sample size of 30 participants is justified, adding to the study's credibility. However, the methodology section could be strengthened by detailing the participant recruitment process to address potential biases. 

● Thank you for bringing this to our attention. We added the following text to the limitations section.

● Further, the convenience sampling recruitment strategy might lead to participants who differ from the broader population. A non-randomized sample may also lead to self-selection bias, where individuals who choose to participate have differing beliefs or values from the wider population. To mitigate those limitations, we will recruit participants using social media and from various school and community-based settings and rural and urban communities to diversify our sample.

Additionally, providing more specific information on the validity and reliability of the instruments used would further enhance the methodology.

● Thank you for the suggestion. We added additional validity and reliability content to the Measures section.

The methodology demonstrates a clear approach to data presentation and analysis, utilizing mixed methods for a comprehensive understanding. However, the lack of preliminary results or discussion of potential challenges in data interpretation leaves readers without insight into expected outcomes, limiting their understanding of the study's potential findings and impact.

The forthcoming discussion section is anticipated to enrich the study by connecting findings to existing literature and exploring the implications for understanding creativity in autistic individuals. The expected acknowledgment of limitations and suggestions for future research reflects a thoughtful approach. However, the absence of the discussion section currently limits a complete evaluation of the study's conclusions and overall impact.

● Thank you for sharing the opportunities for improvement. To address the reviewer’s points, we revised the discussion, limitations, and conclusion sections significantly. 

The conclusion succinctly summarizes the study's objectives and potential contributions, reinforcing its focus by reiterating the research questions and hypotheses. However, due to the lack of results, the conclusion does not provide concrete insights or evidence-based recommendations, which limits its practical applications.

● Thank you for this comment. We significantly revised the Conclusion section to highlight the study's importance and the mixed methods study design. 

● The quantitative outcomes will provide crucial empirical evidence on the effects of an arts-based program on creative self-efficacy, psychological empowerment, and design thinker traits in autistic youth and young adults. Digital artifacts of the partic

---

## [Editor Report · Decision Letter 1]

25 Oct 2024

The impact of art, storytelling, and STEAM-based approaches on creativity development in autistic youth and young adults: A mixed methods study protocol

PONE-D-24-28708R1

Dear Dr. Holt,

We’re pleased to inform you that your manuscript has been judged scientifically suitable for publication and will be formally accepted for publication once it meets all outstanding technical requirements.

Kind regards,

Cho Lee Wong, PhD

Academic Editor

PLOS ONE
---

## [Editor Report · Acceptance letter]

19 Nov 2024

PONE-D-24-28708R1 

PLOS ONE

Dear Dr. Holt, 

I'm pleased to inform you that your manuscript has been deemed suitable for publication in PLOS ONE. Congratulations! Your manuscript is now being handed over to our production team.

Kind regards, 

on behalf of

Dr. Cho Lee Wong 

Academic Editor

PLOS ONE